# The Impact of Long-Term Macrolide Exposure on the Gut Microbiome and Its Implications for Metabolic Control

Jocelyn M. Choo,[a,b] Alyce M. Martin,[c] Steven L. Taylor,[a,b] Emily Sun,[c] Fredrick M. Mobegi,[a,b] Tokuwa Kanno,[d] Alyson Richard,[a,b] Lucy D. Burr,[e,f] Stevie Lingman,[e] Megan Martin,[e] Damien J. Keating,[c,g] A. James Mason,[d] Geraint B. Rogers[a,b]

aMicrobiome and Host Health Program, South Australian Health and Medical Research Institute, Adelaide, South Australia, Australia

bInfection and Immunity, Flinders Health and Medical Research Institute, College of Medicine and Public Health, Flinders University, Bedford Park, South Australia, Australia

cFlinders Health and Medical Research Institute, College of Medicine and Public Health, Flinders University, Adelaide, South Australia, Australia

dInstitute of Pharmaceutical Science, School of Cancer & Pharmaceutical Sciences, King's College London, London, United Kingdom

eDepartment of Respiratory and Sleep Medicine, Mater Adult Hospital, Brisbane, Queensland, Australia

fRespiratory and Infectious Disease Research Group, Mater Research Institute, Brisbane, Queensland, Australia

gNutrition & Metabolism, South Australian Health and Medical Research Institute, Adelaide, South Australia, Australia

**ABSTRACT** Long-term low-dose macrolide therapy is now widely used in the treatment of chronic respiratory diseases for its immune-modulating effects, although the antimicrobial properties of macrolides can also have collateral impacts on the gut microbiome. We investigated whether such treatment altered intestinal commensal microbiology and whether any such changes affected systemic immune and metabolic regulation. In healthy adults exposed to 4 weeks of low-dose erythromycin or azithromycin, as used clinically, we observed consistent shifts in gut microbiome composition, with a reduction in microbial capacity related to carbohydrate metabolism and short-chain fatty acid biosynthesis. These changes were accompanied by alterations in systemic biomarkers relating to immune (interleukin 5 [IL-5], IL-10, monocyte chemoattractant protein 1 [MCP-1]) and metabolic (serotonin [5-HT], C-peptide) homeostasis. Transplantation of erythromycin-exposed murine microbiota into germ-free mice demonstrated that changes in metabolic homeostasis and gastrointestinal motility, but not systemic immune regulation, resulted from changes in intestinal microbiology caused by macrolide treatment. Our findings highlight the potential for long-term low-dose macrolide therapy to influence host physiology via alteration of the gut microbiome.

**IMPORTANCE** Long-term macrolide therapy is widely used in chronic respiratory diseases although its antibacterial activity can also affect the gut microbiota, a key regulator of host physiology. Macrolide-associated studies on the gut microbiota have been limited to short antibiotic courses and have not examined its consequences for host immune and metabolic regulation. This study revealed that long-term macrolides depleted keystone bacteria and impacted host regulation, mediated directly by macrolide activity or indirectly by alterations to the gut microbiota. Understanding these macrolide-associated mechanisms will contribute to identifying the risk of long-term exposure and highlights the importance of targeted therapy for maintenance of the gut microbiota.

**KEYWORDS** azithromycin, erythromycin, gut microbiome, metabolic health, immunoregulation, murine antibiotic model, gnotobiotic model

Address correspondence to Geraint B. Rogers, geraint.rogers@sahmri.com.

The authors declare no conflict of interest.

Long-term, low-dose macrolide therapy is defined as involving a dosage that is lower than, and a duration that exceeds, those typically employed to treat an acute infection (1). Such therapy has been shown to effectively prevent exacerbations in chronic pulmonary disease (2–6). Macrolide therapy appears to provide ongoing benefits, is

well-tolerated, and is considered to carry relatively little risk, and is now widely used to treat asthma, bronchiectasis, chronic obstructive pulmonary disease, bronchiolitis obliterans, chronic rhinosinusitis, cystic fibrosis, organizing pneumonia, and diffuse pan-bronchiolitis (1, 7). However, macrolides are known to have pleiotropic effects, and the potential consequences of long-term continuous exposure to these agents, which can span decades in some cases, is poorly understood.

The agents used in long-term macrolide therapy, such as azithromycin and erythromycin, have the capacity to directly influence many aspects of host physiology, including local and systemic immune regulation (8). Their ability to reduce cellular expression of pro-inflammatory mediators has been demonstrated using *in vitro* and animal models (9). Moreover, through the modulation of intracellular mitogen-activated protein kinase and NF-$\kappa$B pathways, these effects occur across multiple immune cell types (8, 10). Studies in macrolide recipients have similarly shown that macrolides dampen pro-inflammatory airway responses through reduced neutrophilic inflammation in chronic respiratory disease (11, 12), and influence circulating cytokine levels (13–15).

In addition to their influence on immune regulation, macrolide antibiotics have antimicrobial activity against many Gram-positive and Gram-negative bacteria (8). Disruption of the gut microbiome as a result of antibiotic exposure can have a profound impact on host physiology and has been linked to an increased risk of cardiometabolic conditions, including cardiovascular disease, nonalcoholic fatty liver disease, type 2 diabetes, and obesity (16). The gut microbiota and its metabolites can modulate host metabolic regulation by activating enteric cells and promote the release of gut-derived peptides such as serotonin (5-HT) and glucagon-like peptide 1 (GLP-1) (17). These molecules are involved in signaling the enteric nervous system (ENS) to influence gastrointestinal physiological functions such as motility and nutrient absorption, which play an important role in maintaining energy and glucose homeostasis (17). It is therefore possible that long-term macrolide treatment used in chronic lung disease could influence diverse health outcomes, either directly or indirectly via alteration of the gut microbiome.

Changes in intestinal microbiology and microbiome-host interaction are well-recognized consequences of short-term exposure to antibiotics at antimicrobial dosages (18–20). However, the extent to which long-term, low-dose macrolide therapy alters the gut microbiome or influences aspects of host physiology is not known. While addressing these knowledge gaps is essential to our understanding of the potential consequences of macrolide therapy, it presents considerable challenges. Individuals with chronic lung disease typically have high exposure to non-macrolide antibiotics and exhibit markers of chronic systemic inflammation associated with underlying disease and, in some cases, altered intestinal pathophysiology (21). To control these exposures, we undertook a single-blinded, randomized parallel 4-week trial of twice-daily 400 mg erythromycin ethylsuccinate or twice-daily 125 mg azithromycin in healthy adults. This approach enabled us to investigate changes in the gut microbiome and host immune and metabolic homeostasis following exposure to low-dose macrolides and in the absence of potential confounding factors. We then explored the macrolide-associated changes in intestinal microbiology and host physiology that were observed in participants using a preclinical model of long-term erythromycin exposure, and demonstrated causality in these relationships through microbiota transplantation into germ-free mice.

Here, we report the first detailed investigation of the consequences associated with long-term, low-dose macrolide therapy on recipient physiology in healthy adults. We identify direct and microbiome-mediated consequences on host metabolic regulation that could influence long-term disease risks and health outcomes.

## RESULTS

**Low-dose, long-term macrolide exposure is associated with alterations in systemic immune and metabolic markers.** Healthy adults received low-dose erythromycin (ERY) or azithromycin (AZM) for 4 weeks ($n = 10$ per antibiotic group) and effects on systemic immune and metabolic markers were assessed. Because the immuno-modulatory properties of macrolides are recognized, our initial analysis focused on

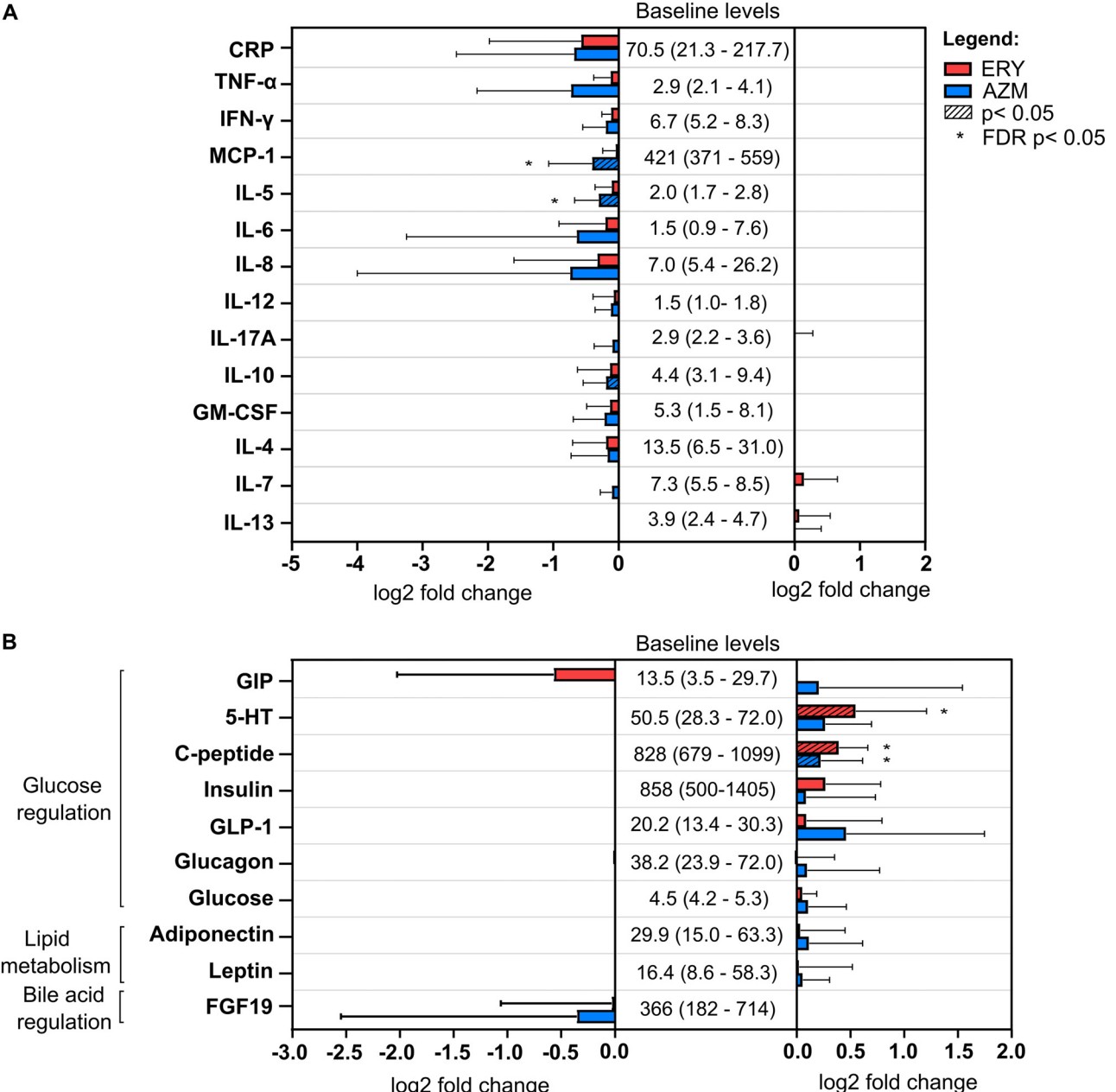

**FIG 1** Changes in serum levels of (A) systemic immune markers and (B) metabolic biomarkers associated with glucose regulation, lipid metabolism and bile acid regulation in erythromycin (ERY) and azithromycin (AZM) groups in humans. Baseline levels of each serum biomarker were based on the mean and standard deviation at baseline across all participants ($n = 20$). All values are in pg/mL except for 5-HT (ng/mL), glucose (mmol/L), adiponectin ($\mu$g/mL), leptin (ng/mL), and C-reactive protein (CRP, ng/mL). Serum biomarkers were measured using commercially available single and multiplex immunoassay panels as described in the Methods section. Bar graph and error bars represent the mean fold change ($\log_2$) and standard deviation of biomarker levels at the end of the antibiotic treatment compared to their respective baseline values. Pairwise comparisons were performed using a linear mixed model (*lme4* v1.1-23 and *lmerTest* v3.1-1 packages in R). Statistical significance ($P < 0.05$) following correction for multiple testing using the false discovery rate (FDR) method is indicated by an asterisk (*).

confirmatory assessment of macrolide-associated changes in a broad panel of immune markers, including cytokines, chemokines, and growth factors, in fasting sera. Exposure to azithromycin was associated with a significant reduction in serum inflammatory markers, including the chemokine monocyte chemoattractant protein 1 (MCP-1) and cytokines interleukin 5 (IL-5; false discovery rate [FDR] $P < 0.05$) and IL-10 ($P < 0.05$) (Fig. 1A, Fig. S1 in the supplemental material). Similar effects were observed with erythromycin, although these did not achieve statistical significance (Fig. 1A, Fig. S1). Tumor necrosis factor alpha (TNF-$\alpha$), granulocyte-macrophage colony-stimulating factor (GM-

CSF), interferon gamma (IFN-$\gamma$), IL-4, IL-6, IL-7, IL-8, IL-12, and IL-17A showed decreasing trends following exposure to either macrolide, but these did not achieve statistical significance. Levels of lipopolysaccharide (LPS), fibroblast growth factor 2 (FGF-2), and the cytokines hepatocyte growth factor (HGF), IL-1$\beta$, and IL-9 were detected in less than 60% of samples in either group and were not analyzed further.

The potential for macrolide exposure to influence human metabolic homeostasis was also explored through an assessment of macrolide-associated changes to serum markers of glucose regulation (gastric inhibitory polypeptide [GIP], 5-HT, C-peptide, insulin, GLP-1, glucagon, glucose, and peptide YY [PYY]), lipid metabolism (adiponectin and leptin), and bile acid regulation (FGF-19) (Fig. 1B, Fig. S2). Baseline levels were within normal ranges for all participants, while PYY was detected in less than 60% of samples in either group. Erythromycin and azithromycin were both associated with significant increases in C-peptide levels, while significant increases in serum levels of 5-HT were observed with erythromycin alone (FDR $P < 0.05$) (Fig. 1B, Fig. S2). The changes in insulin levels following macrolide treatment were positively correlated with those of C-peptide ($r = 0.523$, $P = 0.018$), consistent with the function of C-peptide as a stable biomarker for insulin secretion. Changes in the levels of GIP, insulin, GLP-1, glucagon, glucose, adiponectin, and leptin trended upwards, while a nonsignificant downward trend was observed for FGF-19.

**Macrolide exposure is associated with alteration of gut microbiota composition.** Given the potential for changes in intestinal microbiology to influence host physiology, we explored the impact of low-dose macrolide exposure on the gut microbiome. Macrolide treatment was not associated with significant reductions in fecal bacterial load ($P > 0.05$ for both ERY and AZM) (Table S1). However, changes in gut microbiota alpha diversity were observed with both antibiotics. Specifically, compared to the baseline, treatment was associated with a substantial decline in the number of bacterial taxa detected (ERY, $P = 0.006$; AZM, $P = 0.006$; combined, $P < 0.0001$) and their diversity (ERY, $P = 0.010$; AZM, $P = 0.004$, combined, $P < 0.0001$), but not bacterial evenness (Pielou's evenness; $P > 0.05$) (Fig. S3A to C, respectively). Significant alterations in fecal microbiota composition were also observed following macrolide treatment (ERY: $P = 0.002$, $t = 2.16$; AZM: $P = 0.004$, $t = 1.99$; combined: $P = 0.0002$, pseudo-F = 7.82) (Fig. 2A, Table S2). Notably, the composition of the altered microbiota following macrolide exposure did not differ significantly between the erythromycin and azithromycin groups ($P = 0.826$, $t = 0.88$). The impact of erythromycin and azithromycin was consistent across the participant cohort, with no significant increase in within-group variance (dispersion) compared to the baseline (ERY, $P = 0.70$; AZM, $P = 0.46$; combined, $P = 0.44$).

Both azithromycin and erythromycin were associated with depletion of specific bacterial taxa, including *Bifidobacterium longum*, *Bifidobacterium adolescentis*, *Akkermansia muciniphila*, and *Bilophila wadsworthia* (FDR $P < 0.05$) (Fig. 2B). Conversely, the relative abundances of *Actinomyces johnsonii*, *Eggerthella lenta*, *Ruminococcus gnavus*, and *Coprococcus comes* were significantly increased in both groups (FDR $P < 0.05$). Agent-specific changes in the relative abundance of bacterial taxa were also observed. For example, erythromycin alone was associated with significant reductions in *Ruminococcus bicirculans*, *Roseburia inulinivorans*, and *Actinomyces* sp. ICM47, and increased relative abundance of *Blautia obeum*, *Flavonifractor plautii*, and *Anaeromassibacillus* sp. ($P < 0.05$) (Fig. 2C). In contrast, azithromycin alone was associated with depletion of a *Firmicutes* bacterium CAG110, *Colinsella stercoris*, and *Odoribacter splanchnicus*, and increased relative abundance of *Actinomyces* sp. oral taxon 180 and *Eubacterium ramulus* ($P < 0.05$) (Fig. 2D).

**Macrolide exposure is associated with changes in the functional capacity of the fecal microbiome.** Consistent with changes to taxonomic composition, macrolide exposure was also associated with significant changes in the functional capacity of the fecal microbiome (ERY: $P = 0.014$, $t = 2.31$; AZM: $P = 0.003$, $t = 2.42$; Fig. 3A, Table S3). Of note, reductions in pathways involved in carbohydrate biosynthesis and degradation and glycolysis were observed in both groups, but were more pronounced in the group receiving azithromycin (FDR $P < 0.05$) (Fig. 3B). Reductions in mixed acid fermentation and homolactic fermentation pathways, which are associated with the generation of short-chain fatty acid

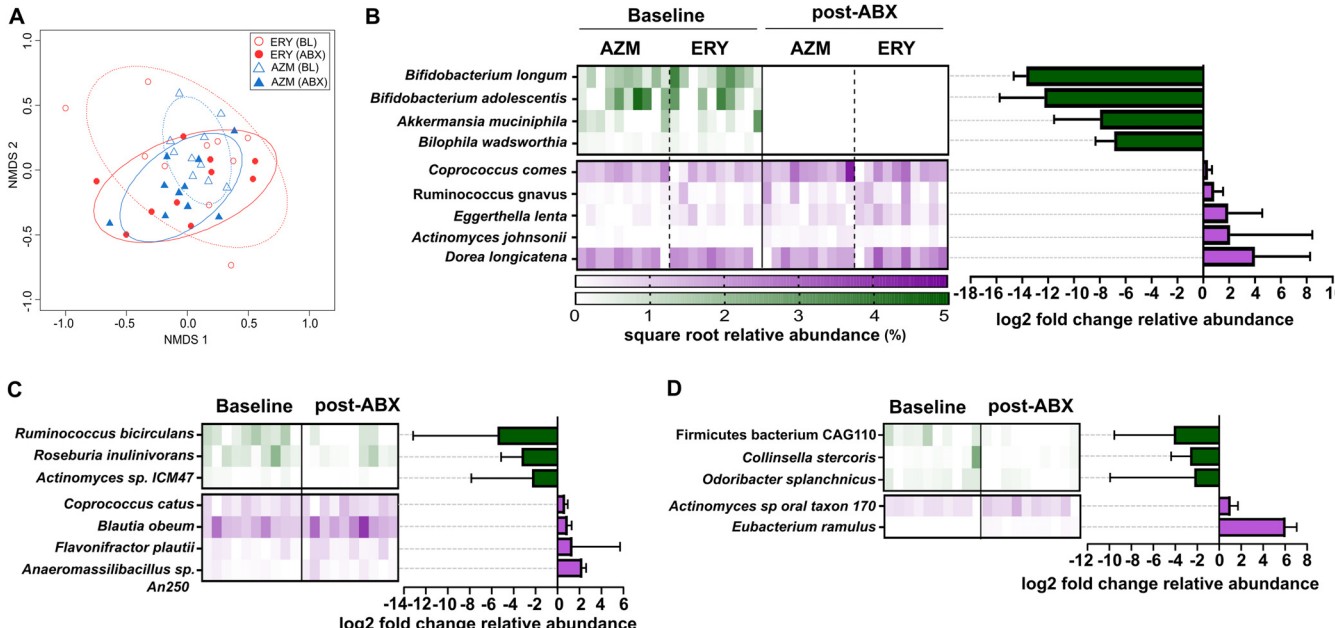

**FIG 2** (A) Nonmetric multidimensional plot based on fecal microbiota before (baseline, BL) and after 4 weeks of erythromycin or azithromycin treatment (ABX). Ellipses for each group represent the standard deviation with 80% confidence limit (dotted and solid lines represent baseline and tretment groups, respectively). (B to D) Microbial species that were significantly altered by both erythromycin and azithromycin treatment in humans, and those that were altered either by (C) erythromycin or (D) azithromycin. Heatmap represents the square root relative abundances of samples within each group; bar graphs represent the (log₂) fold change in the relative abundance of specific taxa between baseline and the end of 4 weeks of macrolide treatment. Bars and error bars depict the median and interquartile ranges. Significance was determined based on the Wilcoxon test at FDR $P < 0.05$ for comparisons involving both erythromycin and azithromycin groups, and $P < 0.05$ for agent-specific comparisons.

(SCFA) precursors acetate and/or lactate as well as pathways involved in the citric acid (TCA) cycle, which utilizes intermediates of fermentation pathways (such as acetyl coenzyme A, oxaloacetate, and citrate), were also observed for both groups ($P < 0.05$) (Fig. 3B).

Of the species which displayed changes in relative abundance following macrolide treatment, a reduction in *B. longum* was a potential principal contributor to many of the functional changes identified (Fig. 3B). These changes included pathways associated with carbohydrate metabolism (e.g., UDP-*N*-acetyl-D-glucosamine biosynthesis I, glycogen degradation I, trehalose degradation V, and sucrose degradation IV), as well as the gamma-glutamyl cycle, which is associated with the biosynthesis and degradation of glutathione, an essential molecule involved in immune function and nutrient metabolism. Shifts in other species, including *B. adolescentis*, *B. wadsworthia*, and *R. bicirculans*, also contributed to altered capacity for carbohydrate metabolism (Fig. 3B). Despite changes in microbiome functional capacity, fecal pH was not altered with treatment (ERY, $P = 0.131$; AZM, $P = 0.193$) (Table S1).

**Changes in host physiology are correlated with altered microbiome functional capacity.** We then explored whether macrolide-associated changes in intestinal microbiology were associated with changes in host systemic immunity and metabolic regulation. Significant relationships were identified, particularly for host markers associated with glucose regulation (Fig. S4). Notably, increases in 5-HT and C-peptide were associated with depletion of *Bifidobacterium*, pathways involved in glucose utilization and synthesis (including gluconeogenesis, glycolysis and the degradation of sugars) as well as fermentation pathways (mixed acid and homolactic fermentation) ($P < 0.05$) that are involved in the synthesis of SCFA precursors such as lactate, acetate, and succinate. Associations between microbiome traits and insulin levels were largely not significant ($P > 0.05$) but followed similar trends to those observed for C-peptide (Fig. S4).

Significant associations between gut microbiota characteristics and host immune markers were largely limited to MCP-1 and GM-CSF (Fig. S4), although no overlapping significant associations between these markers were observed. Changes in IL-5 or IL-10

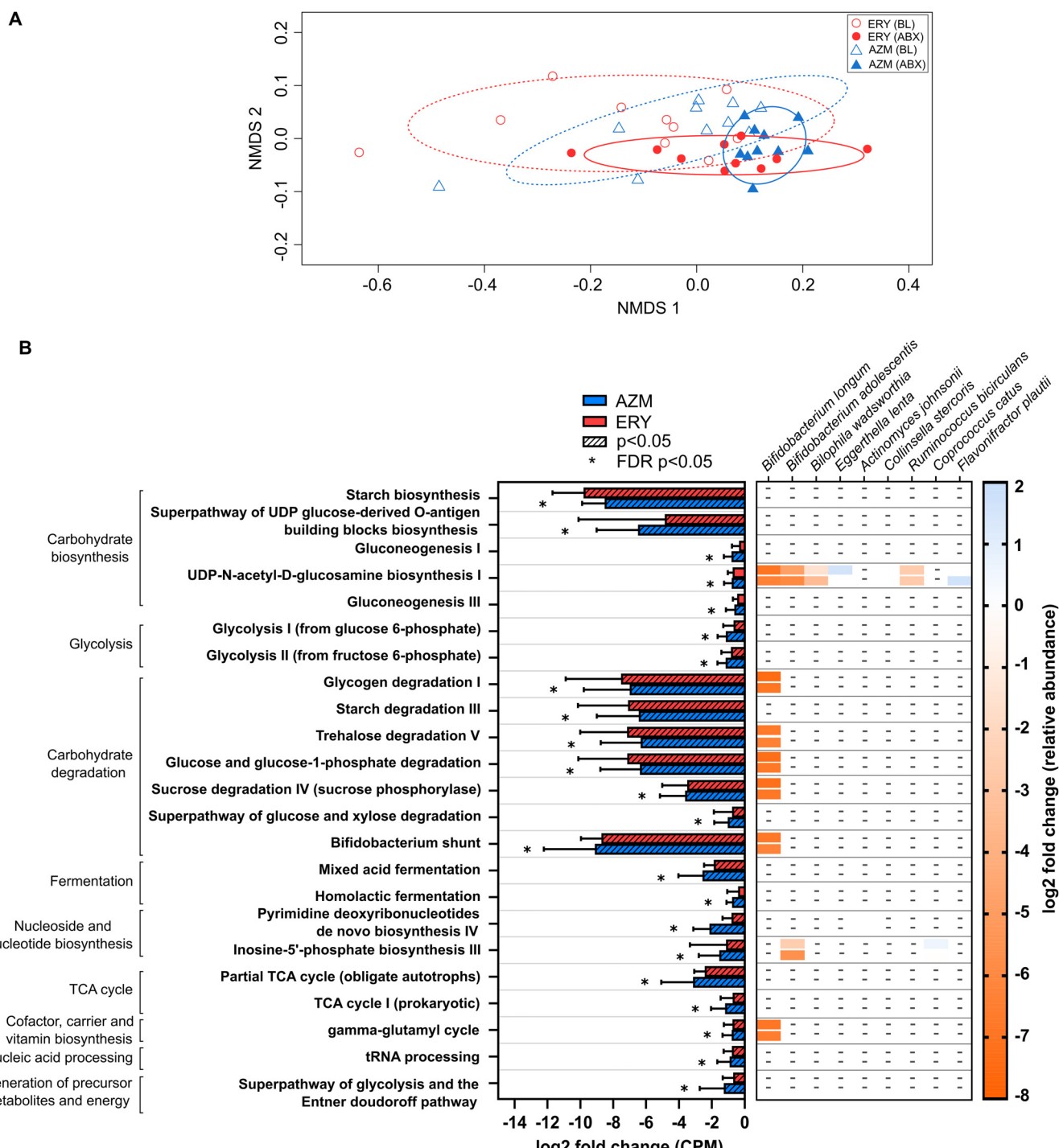

**FIG 3** (A) Nonmetric multidimensional scaling ordination plot of bacterial functional pathway abundance in humans following treatment with azithromycin or erythromycin for 4 weeks. Ellipses for each group represent the standard deviation with 80% confidence limit (dotted and solid lines represent baseline and treatment groups, respectively). (B) Pairwise comparison between baseline and the end of erythromycin (ERY) and azithromycin (AZM) treatment was performed using the Wilcoxon test, and *P* values were corrected for multiple comparisons using the FDR method. Only pathways that were significantly altered by a 1.5-fold change (log$_2$ average fold change > |0.58| and FDR *P* < 0.05) compared to the respective baseline abundance, either in the erythromycin or azithromycin group, are shown. Bars and error bars represent the median and interquartile ranges. Stratification of pathways based on bacterial species that were significantly altered by either erythromycin or azithromycin is shown using a heatmap. None of the taxonomically stratified pathways were significantly altered with treatment. Pathway abundance (counts per million, CPM) were determined using the Human Microbiome Project Unified Metabolic Analysis Network (HUMAnN3) tool and annotated based on the Metacyc Metabolic Pathway database.

were not associated with the microbiome, except for a significant association between IL-10 and the inosine-5′-phosphate biosynthesis pathway (associated with nucleotide biosynthesis). While exploratory, these findings suggest a potential mechanism by which changes in the gut microbiome arising through macrolide exposure might alter host metabolic homeostasis and, to a lesser extent, aspects of immune regulation.

**Erythromycin-associated changes are evident in the murine fecal microbiome.** A murine model of long-term, low-dose erythromycin treatment was used to further investigate relationships between macrolide-associated changes in intestinal microbiology and host metabolism. We first established that changes in fecal microbial community structure in conventional mice following erythromycin treatment (ERY) were largely consistent with those observed in humans. Specifically, there was no significant change in total fecal bacterial load ($P = 0.451$) (Table S4), but erythromycin was associated with a reduction in observed species ($P \leq 0.0001$) and the Shannon diversity index ($P \leq 0.001$) (Fig. S5). These changes were significant compared to the control mice at day 90 ($P \leq 0.05$).

Fecal microbiota composition in erythromycin-treated mice at day 90 differed significantly from baseline (PERMANOVA [permutational multivariate analysis of variance]: $P = 0.0001$, $t = 9.28$) and from the control mice ($P = 0.0001$, $t = 8.72$). This compositional difference was associated with a significant decrease in the relative abundance of anaerobic taxa, particularly those involved in SCFA biosynthesis (*Lactobacillus*, *Faecalibaculum*, *Clostridium sensu stricto* 1, and members of the *Ruminococcaceae* and *Lachnospiraceae* families) and carbohydrate utilization (*Bacteroidales* 24-7 group) in erythromycin-treated mice compared to controls (FDR $P < 0.05$, fold change $> |1.5|$) (Fig. 4A). Notably, a significant increase in the relative abundance of *Akkermansia* with treatment contrasted the depletion observed in humans following macrolide exposure. Additionally, one taxon in the order *Bacteroidales* decreased significantly in erythromycin-treated mice (FDR $P < 0.05$), although no corresponding change was observed in human participants (FDR $P = 0.915$) (Table S2).

The impact of macrolide-associated changes in intestinal microbiology on the production of potential mediators of host-microbiome interaction was then assessed using fecal metabolomics. More than half of the differentially abundant metabolites identified between the groups at day 90 (26 out of 46 detected metabolites) were found to be reduced following erythromycin treatment (FDR $P < 0.05$) (Fig. 4B), consistent with the reduced microbial functional capacity observed in humans. Of note, levels of SCFAs (acetate, propionate, and butyrate) and their precursor compounds, lactate and citrate, were reduced in erythromycin-treated mice compared to the controls. Levels of metabolites implicated in host metabolism, including essential amino acids (including lysine, phenylalanine, methionine, leucine, isoleucine, and valine) and non-essential amino acids (including tyrosine, alanine, and aspartate), as well as metabolites implicated in cardiometabolic diseases and immune responses, such as trimethylamine and histamine, were also reduced following erythromycin treatment (Fig. 4B). Erythromycin-associated effects on fecal metabolites were strongly associated with the changes observed in fecal microbiota composition (distance covariance [dcov] = 0.649, $P < 0.0001$) (Fig. 4C). Again, in keeping with observations in humans, erythromycin was not associated with altered fecal pH, despite changes in fecal microbiota and metabolites (median [interquartile range]: control = 6.9 [6.6 to 7.0], ERY = 6.6 [6.3 to 6.9]; $P = 0.071$).

**The microbiome is necessary for erythromycin-related effects on metabolic homeostasis.** Using conventional and germ-free murine models, we then explored whether associations between treatment-associated changes in the gut microbiome and host biomarkers might be causal. Glucose homeostasis in conventional and germ-free murine models was assessed by an intraperitoneal glucose tolerance test. Low-dose erythromycin treatment for 90 days in conventional mice resulted in a significantly lower glucose area under the curve (AUC, 0 to 120) compared to control mice ($P = 0.004$) (Fig. 5A). In contrast, low-dose erythromycin exposure for 90 days in germ-free mice did not alter glucose homeostasis compared to untreated mice ($P = 0.836$) (Fig. 5B). Finally, transplantation of gut microbiota from erythromycin-treated mice into germ-free recipients resulted in a trend of reduced glucose AUC compared to those transplanted with the control microbiota

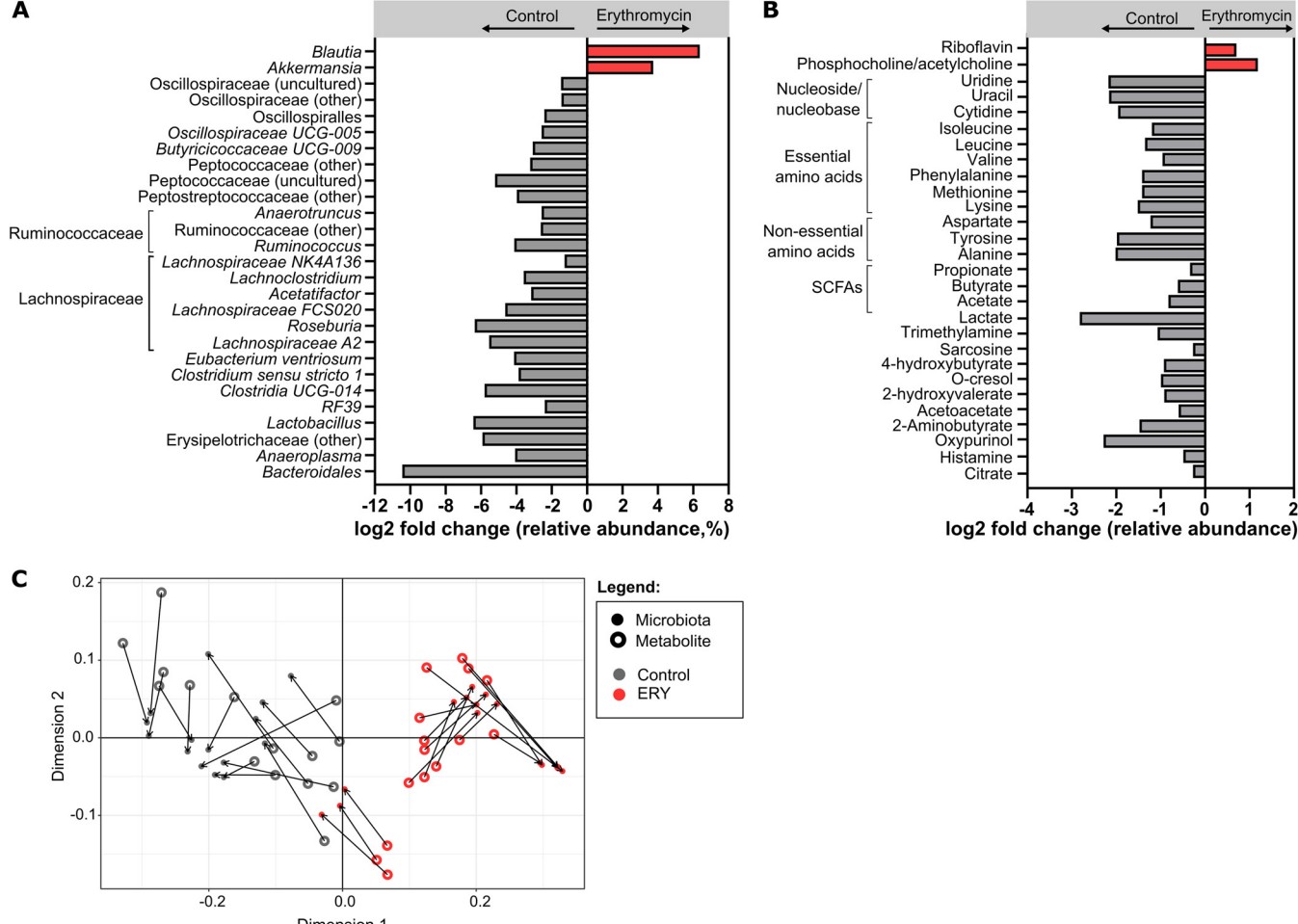

FIG 4 (A) Bacterial taxa and (B) metabolites that were significantly altered in fecal samples of erythromycin-treated mice compared to control mice at day 90. Each bar represents the $\log_2$ average fold change of bacterial taxa (or metabolites) that were significantly higher (in red) or significantly lower (in gray) in erythromycin-treated compared to control mice. (C) Fecal samples of the erythromycin-treated (red symbols) and control mice (gray symbols) were ordinated based on their microbiota composition (closed circles) and metabolite composition (open circles) using Procrustes analysis. The line connecting the closed and open circles represents the distance between microbiota and metabolite samples, respectively, of each mouse. Statistical comparisons of bacterial taxon and metabolite abundances between the groups were performed using the Mann-Whitney test and unpaired $t$ test, respectively. Bacterial taxa that were significantly altered with FDR-adjusted $P < 0.01$ and a fold change of 1.5, as well as significantly altered metabolites (FDR $P < 0.05$), are shown.

(following a similar direction as erythromycin-treated conventional mice), but this did not achieve statistical significance ($P = 0.209$) (Fig. 5C).

The effects of erythromycin on metabolic homeostasis were also assessed based on average respiratory quotient (an indicator of basal metabolic activity) and total body weight (used here as an overarching metabolic indicator). Erythromycin was associated with significantly increased respiratory quotient compared to controls during both day and night cycles ($P < 0.0001$), suggesting effects on nutrient utilization (Fig. S6C). This change was driven primarily by significant increases in the volume of exhaled $CO_2$ during the active night cycle ($P = 0.038$) and was independent of food or water intake and of total energy expenditure ($P > 0.05$) (Fig. S6C). However, body weight did not change significantly with erythromycin treatment in conventional mice ($P = 0.654$) (Fig. S6A). The body weights of germ-free mice treated with erythromycin and germ-free mice transplanted with erythromycin-modified gut microbiota remained unaltered compared to their respective controls ($P > 0.05$) (Fig. S7A and B, respectively).

**Erythromycin influences host metabolism through its alteration of the gut microbiome.** Despite the alterations in glucose homeostasis, serum levels of the metabolic biomarkers 5-HT and C-peptide did not change significantly in response to erythromycin treatment in conventional mice (Fig. 6A). However, germ-free model studies

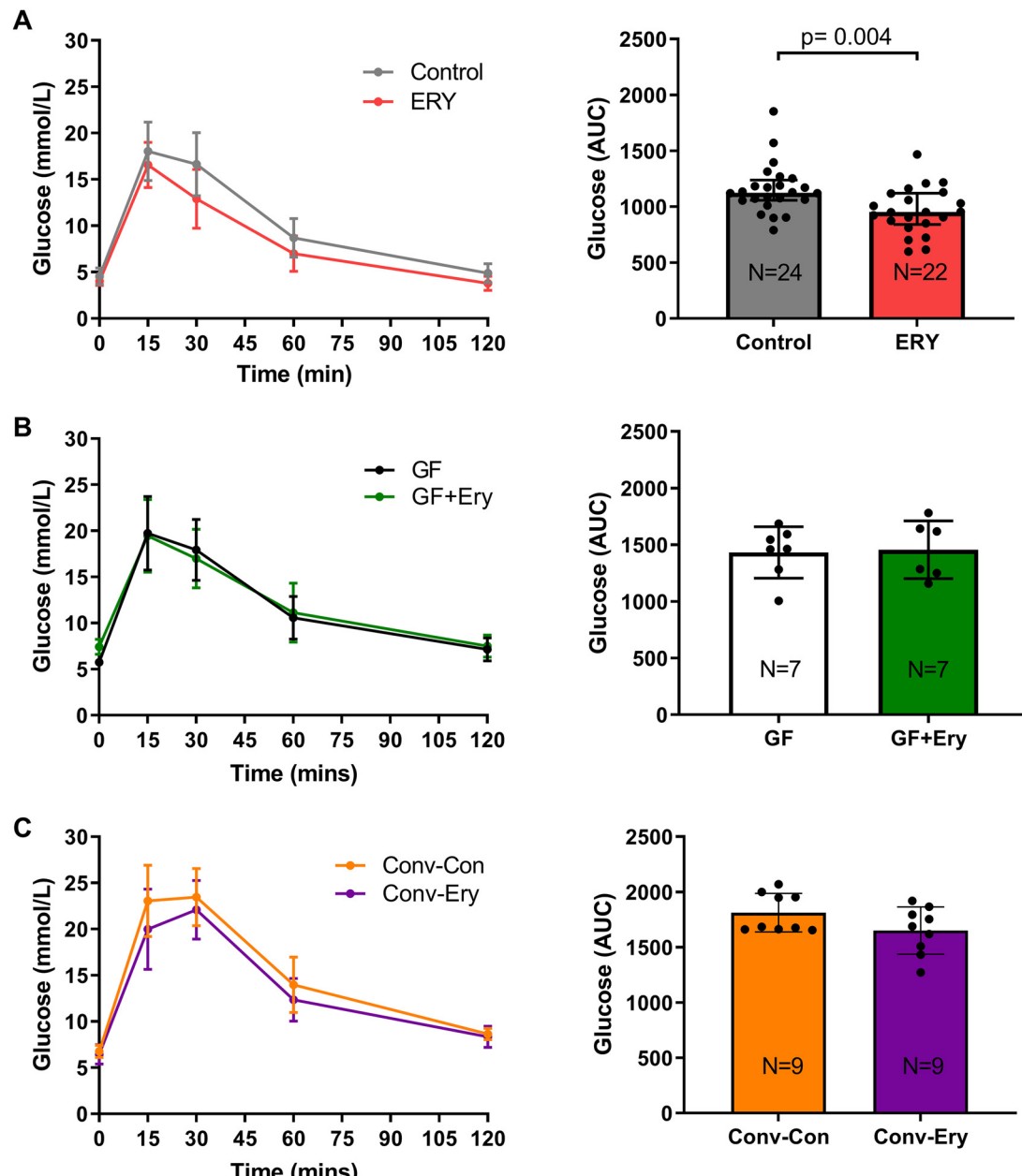

**FIG 5** Blood glucose and the glucose area under the curve (AUC) during a 2-h intraperitoneal glucose tolerance test (IPGTT) at day 90 in (A) mice that received erythromycin (20 mg/kg) or water ($n$ = 22 for ERY and $n$ = 24 for control, respectively), (B) germ-free mice that received erythromycin (20 mg/kg) or water ($n$ = 7 per group), and (C) germ-free mice transplanted with erythromycin-associated microbiota or control microbiota ($n$ = 9 per group). Data are presented as the mean and the error bars represent standard deviation. Statistical comparisons between groups were performed using the Mann-Whitney test, with significance at $P < 0.05$.

indicated that 5-HT levels were significantly modulated, both directly through exposure ($P < 0.001$) and indirectly through microbiota-dependent pathways ($P = 0.002$) (Fig. 6B and C, respectively). These findings suggest that the modulation of 5-HT is not the major driver mediating the changes in glucose homeostasis. No significant direct or microbiota-mediated effects were observed with C-peptide ($P = 0.316$ and 0.107, respectively).

Gut motility is an important regulator of glucose and energy metabolism through its effect on nutrient absorption and is influenced by both intestinal microbiology and prokinetic agents, such as erythromycin. Although direct erythromycin exposure in germ-free mice did not alter gut motility significantly (based on gut transit time) ($P = 0.209$) (Fig. 7A),

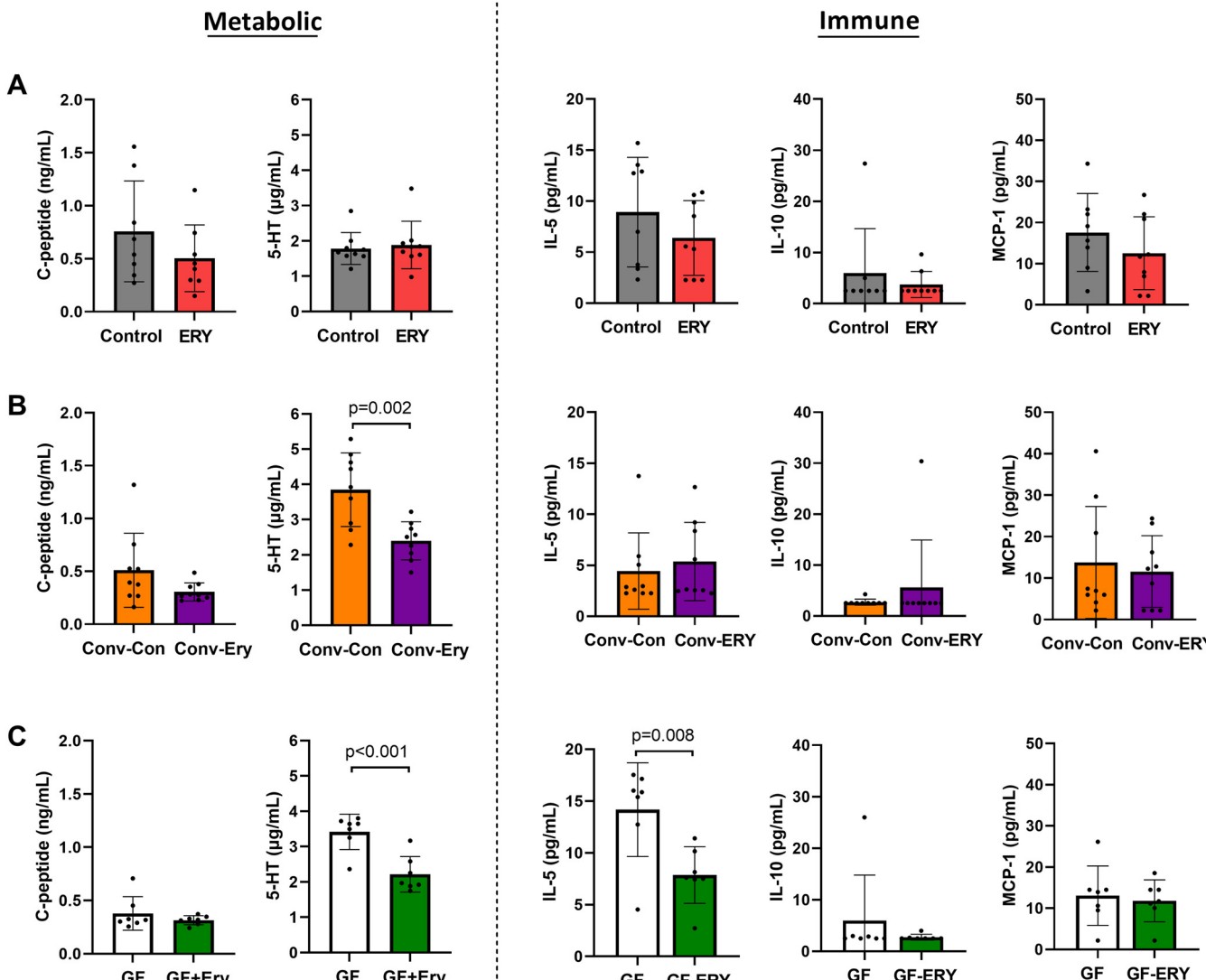

**FIG 6** Serum levels of the metabolic markers C-peptide and 5-HT and the immune markers IL-5, IL-10, and MCP-1 of (A) erythromycin-treated and control mice (*n* = 9 for both groups), (B) germ-free mice and germ-free mice receiving erythromycin treatment (*n* = 7 per group), and (C) germ-free mice colonized with erythromycin-associated or control microbiota (*n* = 9 per group). Serum biomarkers were assessed at the end of the 90-day study. Statistical comparisons were performed using an unpaired *t* test, and significance is set at *P* < 0.05.

transplantation of the erythromycin-associated microbiota into germ-free recipients resulted in significantly prolonged gut transit time compared to the recipients of control microbiota (*P* = 0.0003) (Fig. 7D). Cecum weight followed a similar trend to gut motility, remaining unaltered in germ-free mice directly exposed to erythromycin (*P* = 0.710), but increasing significantly in recipients of erythromycin-associated microbiota compared to recipients of control microbiota (*P* < 0.0001) (Fig. 7B and E, respectively). Notably, cecum weight strongly correlated with gastrointestinal motility in recipients of transplanted microbiota (*r* = 0.89, *P* < 0.0001), a relationship that was absent in non-colonized germ-free mice (*r* = −0.08, *P* = 0.777) (Fig. 7F and C, respectively). Together, these results suggest that erythromycin-associated changes in gastrointestinal physiology primarily arise through alteration of the gut microbial community.

**Evidence of direct erythromycin-associated effects on immune markers.** In addition to the microbiome-mediated impact of erythromycin on glucose homeostasis, we identified evidence of a direct effect of erythromycin on host immune regulation that is independent of macrolide-associated microbiome changes. For example, a significant reduction in IL-5 (*P* = 0.008) was observed in germ-free mice treated with

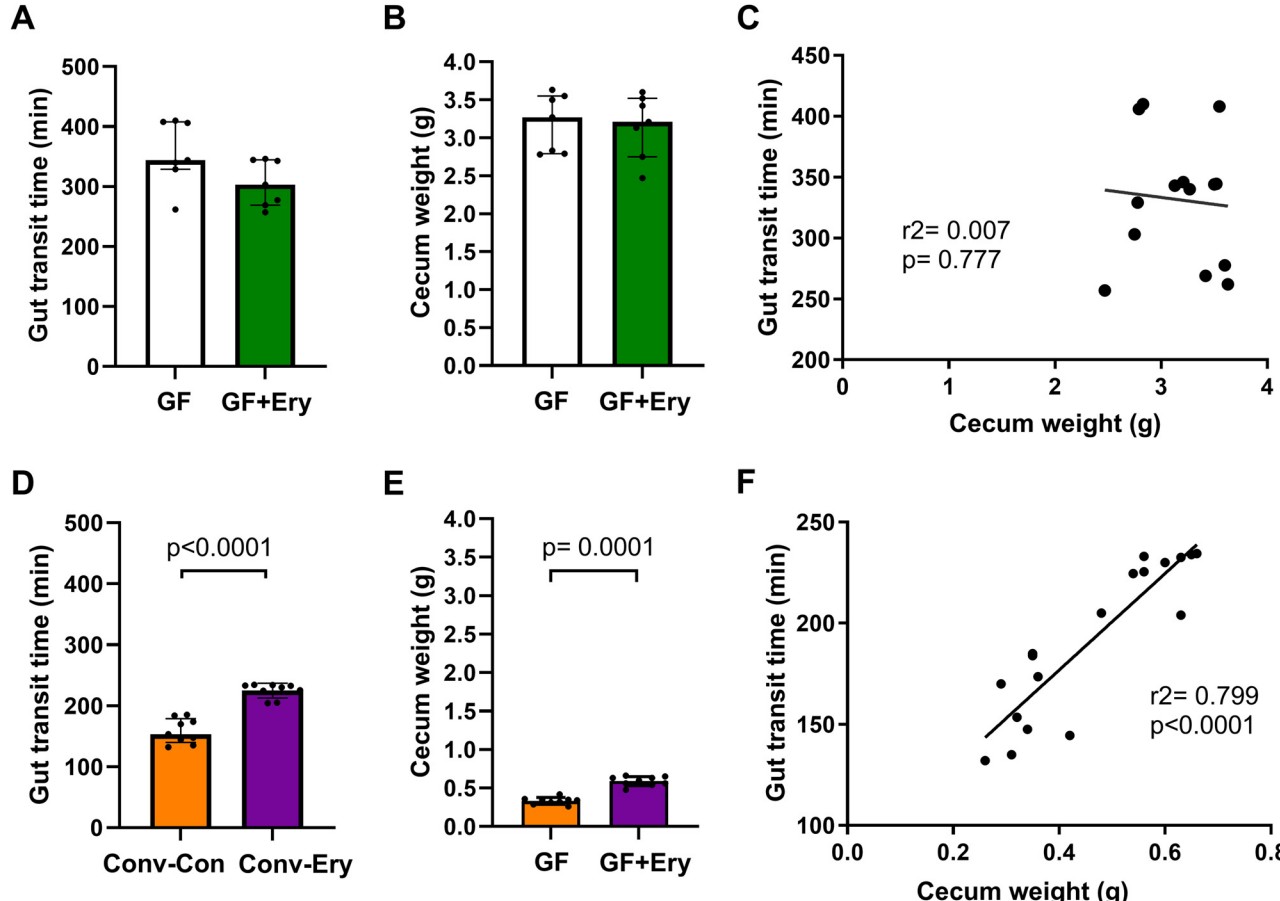

**FIG 7** Assessment of direct effects of erythromycin exposure and erythromycin-associated gut microbiota effects on host physiology. Body weight over the 90-day study was assessed in (A) germ-free mice receiving erythromycin treatment or water and (B) germ-free mice colonized with erythromycin-associated or control microbiota. Cecum weight (C and E) and gut transit times (D and F) were also measured for the respective groups at the end of the 90-day study. Associations between the total gut transit time and cecal weight in (G) germ-free mice receiving erythromycin treatment or water and (H) recipients colonized with control or erythromycin-associated donor microbiota were determined using a linear regression model. Bar graphs and error bars represent the median and interquartile ranges, respectively. Between-group comparisons were statistically analyzed using a Mann-Whitney test, $P < 0.05$.

erythromycin (Fig. 6C), but no difference in IL-5 levels was identified between germ-free recipients of erythromycin-treated or control microbiota (Fig. 6B). Finally, the immune markers IL-10 and MCP-1 remained unaltered in both conventional and germ-free models of erythromycin (Fig. 6).

## DISCUSSION

The efficacy of long-term macrolide therapy at preventing exacerbations in chronic lung disease has led to its increasing clinical use (22, 23). Macrolides carry a risk of QTc interval prolongation and *torsade de pointes* (24), and can result in increased toxicity when co-prescribed with drugs such as statins through inhibition of liver cytochrome P450 enzymes (25). However, with appropriate screening, macrolide therapy is considered to carry minimal risk. Despite this, little is known about the extrapulmonary effects of these agents, which can have pleiotropic effects, either directly or resulting from their impact on the gut microbiome.

Utilizing studies involving healthy adults and preclinical models, we investigated whether macrolide antibiotics at low doses, as used in the management of chronic respiratory conditions, were associated with altered markers of systemic homeostasis. In particular, we focused on an aspect of physiology known to be influenced by altered host-microbiome interactions: metabolic control. In humans, we observed an association between macrolide exposure and alteration of host metabolic markers, changes

that were correlated with changes in gut microbiome composition and function. Our findings suggest that long-term, low-dose macrolide therapy is likely to influence systemic metabolic control in recipients and potentially affect the risk of cardiometabolic disease.

Broadly, we identified associations between macrolide exposure and changes in host markers of immune and metabolic homeostasis in human participants. Modest but significant decreases in serum immune markers IL-5, IL-10, and MCP-1 were observed following macrolide exposure, an effect that was more pronounced with azithromycin. The immuno-modulatory properties of macrolides are well-recognized; for example, asthma patients receiving azithromycin exhibit lower IL-5 expression in CD4$^+$ cells isolated from peripheral blood mononuclear cells (26). The absence of a pronounced anti-inflammatory effect in our study likely reflects the absence of systemic inflammation in healthy participants.

In contrast, exposure to macrolides was associated with increases in host markers of metabolic regulation. In particular, erythromycin significantly increased serum levels of 5-HT and C-peptide, which reflect aspects of glucose regulation. Spore-forming gut microbiota are known to promote 5-HT biosynthesis by colonic enterochrommaffin cells (27). 5-HT, in turn, can affect glucose homeostasis by influencing pancreatic $\beta$-cell production of hormones and regulating lipid metabolism in hepatocytes (28). Erythromycin is a prokinetic and therefore might influence glucose homeostasis by increasing gastrointestinal motility and nutrient absorption. In our investigations in germ-free mice, no increase in motility was observed at the erythromycin dose used, which suggests that erythromycin may influence the host physiology via modulation of the gut microbiota.

Alongside the association between erythromycin and azithromycin and aspects of host physiology, both agents were associated with changes in the composition and functional capacity of the intestinal microbiology. While total bacterial loads were unchanged, both macrolides were associated with significant decreases in bacterial richness and diversity. These findings are similar to the general effects of antibiotic exposure (20), although the macrolide dose used in this study was lower than that prescribed for antibiotic purposes. Treatment was associated with depletion of a number of important commensal anaerobes, including *Bifidobacterium* species (*B. longum* and *B. adolescentis*), while species associated with inflammation, such as *Ruminococcus gnavus* (29), increased following macrolide exposure. These observations align with those reported following acute azithromycin (30–32) or erythromycin therapy (33), and are consistent with the broad-spectrum activity of macrolides when administered at antimicrobial doses (30, 32).

In relation to functional capacity, the abundance of microbial pathways associated with energy metabolism such as glycolysis, carbohydrate biosynthesis and degradation, the TCA cycle, and fermentation were significantly reduced following macrolide exposure in subjects. Parallel investigations in mice revealed macrolide-associated changes in carbohydrate metabolism pathways (UDP-*N*-acetyl-D-glucosamine biosynthesis I and *Bifidobacterium* shunt) and fermentation pathways (fermentation of sugars to acetate and lactate). Direct interrogation of fecal metabolites in mice further substantiated the predicted functional alterations, indicating decreased abundance of SCFAs. These changes are consistent with the observed impact of erythromycin on SCFA-producing taxa, including members of the *Ruminococcaceae* and *Lachnospiraceae* families that convert acetate and lactate to butyrate (34).

To further explore the potential relationship between the microbiome and macrolide-associated shifts in host physiology, including the extent to which the associations observed in humans might be causal, we utilized germ-free mice and mice transplanted with a macrolide-exposed microbiota. These analyses demonstrated that macrolide exposure can influence host metabolic physiology via its impact on the gut microbiome. Specifically, analysis in conventional mice, performed using a glucose tolerance test and the metabolic cage system, indicated a relationship between macrolide exposure and metabolic homeostasis. Alterations in glucose regulation in macrolide-exposed mice, as reflected by a decrease in glucose AUC, were observed concomitantly with increased respiratory quotient in both the resting and active states; these effects are consistent with changes in energy homeostasis (35–37). A similar but modest trend of lower glucose

AUC levels was observed in germ-free mice transplanted with erythromycin-treated microbiota, but not in germ-free mice directly treated with erythromycin. These findings support microbiome-mediation of the observed relationship between erythromycin exposure and glucose homeostasis. In contrast, reduced levels of the pro-inflammatory cytokine IL-5 were associated with direct erythromycin activity and were unaffected by manipulation of gut microbiota composition, suggesting direct mediatory effects of macrolides on immune regulation, consistent with the established anti-inflammatory properties of both erythromycin (38) and azithromycin (26).

Our analysis of germ-free models also indicated that microbiome-dependent effects were the principal mediators of altered gut motility. Gut motility, which plays an important role in glucose homeostasis, is regulated through the contractile activity of the gastrointestinal tract by the enteric nervous system. Microbial components, such as LPS, and microbial metabolites, including products of carbohydrate fermentation, are involved in signaling of the ENS (17, 39). Additionally, gut hormones, including 5-HT, are also known to play a role in ENS signaling (17). Although our germ-free mice studies showed that 5-HT can be significantly modulated by the microbiome and by erythromycin exposure, host gastrointestinal transit was significantly regulated by the gut microbiome. Together, these findings indicate that the effects of erythromycin on the microbiome, rather than its direct effects on the host, are stronger modulators of gastrointestinal physiology and may contribute to metabolic homeostasis.

The changes in gut microbiology and associated alterations to aspects of host physiology that were observed with low-dose, long-term macrolide exposure represent a potential source of unintended treatment effects. Our findings suggest that ongoing macrolide therapy for chronic respiratory disease could have significant direct impacts on systemic immunity and microbiome-mediated changes in metabolic homeostasis. Because these effects have the potential to substantially influence long-term metabolic health outcomes, but may only become evident over a long time period and in relation to diverse morbidities, longitudinal assessments of large cohorts, including matched placebo controls, are required.

Our study had limitations that should be considered. Using healthy subjects enabled us to assess the impact of macrolides in the absence of potential confounders of disease, including lung disease and associated therapies. However, the effects of macrolides on systemic physiology might differ where homeostasis is already disrupted by disease-related factors. Although we examined the effects of two macrolides commonly used in the treatment of chronic lung disease, these effects may differ from those of other macrolides. External factors such as physical activity, diet, and stress can influence the gut microbiome, host metabolism, and inflammatory markers (40). Our longitudinal study enabled paired assessment within subjects to minimize the variation associated with such exposures, which can be large between individuals. However, because a placebo control group was not included, the contributions of natural variation in the assessed markers over time, placebo effects on host physiological markers, or behavioral changes due to study participation (Hawthorne effect), to masking of macrolide-associated effects cannot be determined. Additionally, our assessment of direct effects of macrolides on host physiology was not exhaustive. For example, macrolide-related impairment of autophagic flux (41) or interactions with other drugs (42) may impact host physiology. Both humans and mice were investigated here, which have differences in their underlying gut microbiology, physiological processes, and drug metabolism. Indeed, while we found that both humans and mice displayed alterations in host homeostasis and the functional capacity of the gut microbiota in response to macrolides, there were specific taxa and markers that did not align. For example, *A. muciniphila* increased following erythromycin treatment in mice, while it decreased in humans. These effects were consistent with previous studies and may reflect specific responses of *Akkermansia* to each respective model (18, 31). Finally, our murine studies utilized female mice only and therefore may not have captured differences in gut microbiota and host metabolic or immune relationships that arise due to sex effects (43, 44).

**TABLE 1** Participant demographics[a]

| Characteristic | Erythromycin (n = 10) | Azithromycin (n = 10) | P |
|---|---|---|---|
| Age (yrs) | 36 ± 12 | 40 ± 15 | 0.113 |
| Sex (female), n (%) | 9 (90%) | 6 (60%) | 0.303 |
| BMI (kg/m²) | 27.7 ± 5.9 | 28.5 ± 5.1 | 0.727 |
| QTc (ms), median (IQR) | 425 (421−431) | 411 (392–430) | 0.147 |
| Heart rate (beats per minute) | 76 ± 13 | 72 ± 12 | 0.575 |
| Other medications, dosing regimen | | | |
| Prochlorperazine | 25 mg/day orally, 3 days (n = 1) | - | - |
| Metoclopramide | - | 10 mg/day orally, 1 day (n = 1) | - |

[a]BMI, body mass index; IQR, interquartile range. Data are represented as means ± standard deviation unless stated otherwise. P values were calculated using a t test (age, BMI and heart rate), Mann-Whitney test (QTc), or Fisher's exact test (sex), according to data characteristics.

## MATERIALS AND METHODS

**Participant recruitment and sample collection in human studies.** Healthy adults received a 4-week course of twice-daily oral 400 mg erythromycin ethylsuccinate or twice-daily oral 125 mg azithromycin (n = 10 per antibiotic group) (Table 1) (45). Each dose is equivalent to those received by patients prescribed low-dose, long-term macrolide therapy. Participants were randomized in a parallel design and blinded to the antibiotics received. All participants provided full written informed consent. The trial was approved by institutional ethics committees (HREC/15/MHS/41) and registered with the Australian and New Zealand Clinical Trials Registry (no. ANZCTR12617000278336). Fresh fecal samples were collected immediately prior to the start of antibiotic treatment (baseline) and at the end of the 4-week antibiotic treatment and stored at −80°C. Fasting (>6 h) sera were obtained from blood samples collected at corresponding time points. Participants were instructed to maintain their habitual diet during the study period. Participants had not received antibiotics during the preceding 3 months or macrolides during the preceding 12 months. Full eligibility criteria are provided in the supplemental material. Baseline demographics of participants are detailed in Table 1.

**Antibiotic exposure in mice.** Because azithromycin is insoluble in drinking water, only erythromycin was investigated in murine models. C57BL/6 female mice (7 to 8 weeks old) were randomized to receive 20 mg/kg erythromycin ethylsuccinate in drinking water, or plain water (n = 24 per group, three to five mice per cage, assigned into six cages), for 90 days. Dosage calculation of erythromycin ethylsuccinate is detailed in the supplemental material. Fresh fecal pellets were collected prior to antibiotic treatment and on day 90 (supplemental information). Mice had ad libitum access to water and food (Teklad Global 18% Protein Rodent Diet, Envigo, Huntington, United Kingdom), and were bred and maintained at the South Australian Health and Medical Research Institute (SAHMRI) Bioresources animal facility (South Australia, Australia). Mice studies and procedures performed were approved by the SAHMRI Animal Ethics Committee (under SAM133, SAM269, and SAM378).

**Antibiotic exposure and microbiota transplantation in germ-free mice.** Germ-free C57BL/6 female mice (6 to 7 weeks old) were randomized into IsoP cages (Techniplast, Italy) containing three mice per cage, and assigned into four groups (n = 7 to 9 mice per group). Germ-free mice received either plain drinking water or water containing erythromycin ethylsuccinate (equivalent to 20 mg/kg) for 90 days. Engraftment of germ-free mice with donor control or erythromycin-disrupted microbiota was performed using pooled cecum material prepared from three donor mice from the respective groups, as described previously (46, 47). All mice had ad libitum access to water and autoclaved food (Teklad Global 18% Protein Rodent Diet).

**Measurements of host physiology.** Fecal pH of human fecal samples (48) and mouse fecal pellets (46) was measured using a FE20 FiveEasy pH meter (Mettler-Toledo AG, Schwerzenbach, Switzerland). Gastrointestinal transit time in mice was assessed using carmine red dye (3% [wt/vol] solution in 0.5% methylcellulose) (Sigma-Aldrich, St. Louis, USA), as previously described (49). Mouse cecum weight was recorded at the end of the study (supplemental material).

**Fecal DNA extraction and microbial profiling.** Bacterial DNA from human fecal samples and mouse fecal pellets was extracted using a DNeasy PowerLyzer PowerSoil kit according to the manufacturer's instructions (Qiagen, Hilden, Germany), with modifications (50). Quantitation of the total bacterial load (by targeting the 16S rRNA gene) were performed using a SYBR Green-based assay, as previously described (51).

Shotgun metagenomic sequencing libraries was performed on all extracted human fecal DNA using a TruSeq Nano DNA Library Prep kit and paired-end sequencing (150 bp) on a NovaSeq 6000 system (Illumina, San Diego, USA). Sequence read output and quality filtering parameters are described in the supplemental material. Samples used for downstream analysis according to previous parameters (52), had 22.3 ± 2.9 million quality-filtered reads of at least 138 nucleotides and a Q30 score of ≥20. Species-level microbial profiling was performed using MetaPhlAn3, and functional profiling of gene families and metabolic pathway abundances was performed using HUMAnN3, tools within the bioBakery workflow (53).

Extracted DNA from mouse fecal pellets was used for 16S rRNA amplicon sequencing of the V4 hypervariable region (50). Amplicon libraries were indexed and paired-end sequenced using a MiSeq v3

kit (2 × 300 bp) using the Illumina MiSeq platform at the South Australian Genomics Centre (SAGC) in Adelaide (Australia). Sequence data were subsampled to 4,459 reads prior to downstream analysis. Computational analysis of paired-end reads to measure alpha diversity (observed species, Pielou evenness, Shannon diversity H′), beta diversity (weighted UniFrac distances), and bacterial relative abundance were performed using the QIIME2 platform (54), as described previously (46).

**Serum protein quantification.** Serum concentrations of immune-related biomarkers, including cytokines and chemokines (TNF-$\alpha$, IFN-$\gamma$, GM-CSF, MCP-1, IL-1$\beta$, IL-4, IL-5, IL-6, IL-7, IL-8, IL-9, IL-10, IL-12, IL-13, IL-17A), growth factors (HGF, FGF-2), a marker of inflammation (C-reactive protein [CRP]), and lipopolysaccharide (LPS); as well as metabolic markers, including hormones and molecules associated with host glucose metabolism (GIP, 5-HT, C-peptide, insulin, GLP-1, glucose, glucagon, and PYY), lipid metabolism (adiponectin and leptin), and bile acid homeostasis (FGF-19), were quantified using a combination of commercially available multiplex immunoassay panels (Milliplex) and enzyme-linked immunosorbent assays (ELISA). Assays were performed according to the manufacturer's instructions with modifications (supplemental material).

**Fecal metabolome analysis.** Murine fecal metabolome characteristics were determined by proton nuclear magnetic resonance ($^1$H NMR), as described previously (55). Fecal metabolome analysis was performed with NMR spectral intensities subjected to probabilistic quotient normalization and Pareto scaling (56).

**Glucose tolerance test.** Mice were fasted for 12 to 14 h by removing the feeding tray and housing in fresh bedding with fasting trays. All mice had *ad libitum* access to plain drinking water or drinking water containing erythromycin ethylsuccinate (equivalent to 20 mg/kg) during the fasting period. Glucose (2 g/kg of body mass, dissolved in sterile 0.9% sodium chloride) was administered by intraperitoneal injection and blood glucose was measured from the tail tip using a glucometer (Abbott Freestyle Freedom Lite, Australia). Blood glucose measurements were performed at 0 (basal level), 15, 30, 60, and 120 min after glucose administration.

**Metabolic and behavioral monitoring in mice.** Mice were housed individually in a Promethion Metabolic cage system (Sable Systems International, NV, USA) ($n = 3$ per group). Following a 24-h acclimatization period, metabolic and behavioral information was recorded over two consecutive day-and-night cycles.

**Statistical analysis.** Normality of the data distribution was determined using the Shapiro-Wilk test. Comparisons between paired samples were performed using a *t* test (parametric data) or a Wilcoxon test (nonparametric data). Unpaired samples were analyzed using an unpaired *t* test (parametric data) or a Mann-Whitney test (nonparametric data). Paired comparisons of pre- and post-antibiotic human fecal samples were performed to minimize diet-associated effects on study outcomes. Between-group differences in microbiota composition were assessed by PERMANOVA (57), on distance matrices using PRIMER7 (PRIMER-E, Plymouth), based on Bray Curtis distances (shotgun metagenomics data) and weighted UniFrac distances for genus-level relative abundances (16S rRNA amplicon sequence data). Murine microbial diversity analyses were performed using a mixed model with cage as a random factor. Correlations between the distance matrices of murine gut microbiota and metabolome data were assessed by testing for distance covariances using the R package *dcov* (v0.1.1). For serum biomarker analysis, only those whose levels reached above the detection threshold for ≥60% of samples across the erythromycin or azithromycin groups were analyzed. Pairwise comparisons of serum biomarker levels were performed using a linear mixed model (*lme4* v1.1-23 and *lmerTest* v3.1-1) in R. Correlation analysis between microbiome characteristics and host biomarkers was performed using the R package *rmcorr* (v0.5.4) for repeated measures correlation. Assessments of changes to the microbiome, and host immune and metabolic markers in response to macrolides, were corrected for multiple testing using the false discovery rate method, while exploratory correlation analyses between changes were uncorrected.

**Data availability.** The data sets generated during and/or analyzed during the current study are accessible in the NCBI Sequence Read Archive repository under accession no. PRJNA851177 (for shotgun metagenomic sequence data), or PRJNA851193 and PRJNA592263 (for 16S rRNA amplicon sequence data).

## SUPPLEMENTAL MATERIAL

Supplemental material is available online only.
**SUPPLEMENTAL FILE 1**, PDF file, 1.7 MB

## ACKNOWLEDGMENTS

We thank Amanda Page for assistance on the Promethion Metabolic cage studies, and Samay Trec and Mariah Turelli for their assistance in germ-free murine studies. NMR experiments described in this paper were carried out at the Centre for Biomolecular Spectroscopy, King's College London, on instruments acquired with a Multi-User Equipment Grant from the Wellcome Trust and an Infrastructure Grant from the British Heart Foundation. We thank Andrew Atkinson for helping perform NMR experiments.

J.M.C., L.D.B., and G.B.R. conceptualized the study; S.L. and M.M. contributed to acquisition of samples and data for the human study; J.M.C. and A.R. contributed to acquisition of samples and data for murine studies; J.M.C., A.M., E.S., F.M.M., and T.K. conducted formal analysis and interpretation of data; J.M.C., S.L.T., L.D.B., D.J.K., A.J.M.,

and G.B.R. contributed to the preparation and revision of the manuscript. All authors reviewed and approved the final manuscript.

We declare that we have no competing interests.

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
