## [Reviewer comments · Microbiology Spectrum]

Microbiology Spectrum

The impact of long-term macrolide exposure on the gut microbiome and its implications for metabolic control

Jocelyn Choo, Alyce Martin, Steven Taylor, Emily Sun, Fredrick Mobegi, Tokuya Kanno, Alyson Richard, Lucy Burr, Stevie Lingman, Megan Martin, Damien Keating, A. James Mason, and Geraint Rogers

Corresponding Author(s): Jocelyn Choo, South Australian Health and Medical Research Institute Limited

Review Timeline:

Submission Date:	February 24, 2023
Editorial Decision:	May 1, 2023
Revision Received:	May 22, 2023
Accepted:	June 1, 2023

Editor: Ana Weil

Reviewer(s): The reviewers have opted to remain anonymous.

Transaction Report:

DOI: <https://doi.org/10.1128/spectrum.00831-23>

April 30, 2023

Dr. Jocelyn M Choo
South Australian Health and Medical Research Institute Limited
Adelaide, South Australia 5000
Australia

Re: Spectrum00831-23 (The impact of long-term macrolide exposure on the gut microbiome and its implications for metabolic control)

Dear Dr. Jocelyn M

Link Not Available

The ASM Journals program strives for constant improvement in our submission and publication process. Please tell us how we can improve your experience by taking this quick Author Survey

Please include the limitations to your conclusions drawn from the human data in the context of no placebo group. Heatmaps also need legends.

Sincerely,

Ana Weil

Journals Department
Reviewer comments:

Reviewer #1 (Comments for the Author):

Choo et al. conducted an intriguing translational study investigating the effects of macrolide-class antibiotics on host metabolism and inflammation. The study consisted of two parts. The first part was a small (20 subjects), randomized trial comparing the impact of azithromycin and erythromycin in healthy volunteers, which has its limitations, including the absence of placebo control, unblinded participants, and the concurrent use of anti-nausea medications in 10% of the population (Prochlorperazine in the erythromycin group and metoclopramide in the azithromycin group). In this part, the authors measured microbiome changes and markers of metabolism and inflammation. They performed an exploratory analysis of the changes in the microbiome's functional capacity associated with antibiotic treatment.

They concluded that azithromycin and erythromycin altered the microbiome, host metabolism, and inflammation.

The second study was a murine model looking at erythromycin's influence on the gut microbiome, host metabolism, and host inflammation. The authors treated germ-free and conventional mice with erythromycin, measured changes in metabolism and inflammation, and found that erythromycin treatment caused impaired glucose tolerance and changes in inflammation. They then transplanted the fecal microbiome of antibiotic-treated and untreated mice into germ-free mice and found that fecal transplantation altered host metabolism but not inflammation. They concluded that changes in inflammation associated with erythromycin treatment directly affected the medication, while metabolic and inflammatory changes were mediated by the microbiome.

This is a compelling study. The murine model is well-designed, yielding fascinating results. However, I have significant concerns regarding the human randomized trial. The authors justify the lack of blinding and placebo control by stating, "This was felt acceptable as all outcomes were microbiological, not patient-orientated, and therefore not subject to bias." (Page 2, paragraph 1 in the supplement). I must express my strong disagreement with this assertion. The human data are an adjunct to the murine model and are informative but insufficient to establish causality. In my view, the study's real novelty is the murine model.

Major concerns:

1. Hawthorne effect: Since all participants in the human study knew their treatment status, they may modify their behavior accordingly, such as adhering to a healthier diet or exercising more, or any other variety of behavioral changes. These behavioral changes could influence the gut microbiome metabolism and inflammatory markers independently of the antibiotics. The authors claim, "Participants maintained their habitual diet during the study period." How was this measured, and what data support this claim? How much variation was there in the participant's diet at baseline? What about exercise regimen, sleeping cycle, or psychological stress, all of which are associated with gut microbiome changes? Erythromycin and azithromycin frequently cause nausea, and it appears that one participant in each treatment arm became nauseous and required the use of either prochlorperazine or metoclopramide. Surely, these two participants had at least some changes in their dietary habits if they became nauseous. Without blinding or placebo controls, a whole host of behavioral changes must be measured, precisely quantified, and modeled to show that these aren't confounding results. Unblinded studies will always be limited by unmeasured confounding.
2. Placebo effect: Placebo control has been shown to change systemic inflammation in studies of auto-immune disorders, including specific markers that the authors focused on, such as CRP (PMID: 32936297), and also influence host metabolism in other randomized trials (PMID: 36193169). Separating the antibiotic's impact from the placebo effect is impossible without a placebo control.
3. Regression to the mean: Some changes observed in the gut microbiome or inflammatory markers could be due to natural fluctuations over time rather than a direct effect of the antibiotics. Without a control group, it is impossible to determine the extent of this phenomenon.
4. That said, human data has merit, as it still has the advantage of temporality, and before/after comparisons can still be made. Still, this design and type of analysis are not quite at the evidence level to prove causality, and the authors should soften this language to "associated with."
5. Causal claims can be made when comparing azithromycin to erythromycin. The fact that there was no significant difference in gut microbiome changes between the groups is reassuring validation for the author's choice to study erythromycin alone for the murine model (given the inability to administer azithromycin through the water in mice). It is likely representative of azithromycin as well.
6. The supplemental table 1 describing the cohort should be moved to the main manuscript since part of its function is to address some of the limitations of confounding and bias and detect any measurable confounding between groups.

Minor concerns:

1. Consider using ellipses rather than "star patterns" of lines in the NMDS plots.
2. Just a point of clarification. In the supplemental methods, the authors state, "Faecal pellet collection before antibiotic treatment (20mg/kg erythromycin ethylsuccinate, supplemented in drinking water) and at day 90 was performed by placing mice into individual clean cages." (supplemental methods, page 2, first paragraph in Mice Studies Section). Does this mean conventional mice were co-housed and only briefly removed and placed into individual clean cages for drug administration? I'm presuming that the authors took measures to control for the cage effect (PMID: 35545042), but please add a little more detail here to explain how the cage effect was accounted for. I think this is well described for germ-free mice in lines 440-448, but more detail would also be appreciated for the conventional mice.

Reviewer #2 (Comments for the Author):

This is an interesting study and meaningful for this research area. The manuscript was drafted well, then I only several main concerns as followings:

1. Please the authors clarify the rationals of sampling sizes used in the study.
2. Is it plausible to use parametric tests on microbial compositions comparisons?
3. What is the scale for those heatmaps in Figure 2.
4. Legends of Figure 3 should be specified for different panels.
5. How the authors to exclude the confounding factors e.g. diet.
5. All code used in the study is recommended strongly to deposit on github or other.
6. I also suggest authors to make a schematic figure to conclude the findings from the study at last.

Staff Comments:

Preparing Revision Guidelines

Please return the manuscript within 60 days; if you cannot complete the modification within this time period, please contact me. If you do not wish to modify the manuscript and prefer to submit it to another journal, please notify me of your decision immediately so that the manuscript may be formally withdrawn from consideration by Microbiology Spectrum.

Choo et al. conducted an intriguing translational study investigating the effects of macrolide-class antibiotics on host metabolism and inflammation. The study consisted of two parts.

The first part was a small (20 subjects), randomized trial comparing the impact of azithromycin and erythromycin in healthy volunteers, which has its limitations, including the absence of placebo control, unblinded participants, and the concurrent use of anti-nausea medications in 10% of the population (Prochlorperazine in the erythromycin group and metoclopramide in the azithromycin group). In this part, the authors measured microbiome changes and markers of metabolism and inflammation. They performed an exploratory analysis of the changes in the microbiome's functional capacity associated with antibiotic treatment. They concluded that azithromycin and erythromycin altered the microbiome, host metabolism, and inflammation.

The second study was a murine model looking at erythromycin's influence on the gut microbiome, host metabolism, and host inflammation. The authors treated germ-free and conventional mice with erythromycin, measured changes in metabolism and inflammation, and found that erythromycin treatment caused impaired glucose tolerance and changes in inflammation. They then transplanted the fecal microbiome of antibiotic-treated and untreated mice into germ-free mice and found that fecal transplantation altered host metabolism but not inflammation. They concluded that changes in inflammation associated with erythromycin treatment directly affected the medication, while metabolic and inflammatory changes were mediated by the microbiome.

This is a compelling study. The murine model is well-designed, yielding fascinating results. However, I have significant concerns regarding the human randomized trial. The authors justify the lack of blinding and placebo control by stating, "This was felt acceptable as all outcomes were microbiological, not patient-orientated, and therefore not subject to bias." (Page 2, paragraph 1 in the supplement). I must express my strong disagreement with this assertion. The human data are an adjunct to the murine model and are informative but insufficient to establish causality. In my view, the study's real novelty is the murine model.

Major concerns:

1. Hawthorne effect: Since all participants in the human study knew their treatment status, they may modify their behavior accordingly, such as adhering to a healthier diet or exercising more, or any other variety of behavioral changes. These behavioral changes could influence the gut microbiome metabolism and inflammatory markers independently of the antibiotics. The authors claim, "Participants maintained their habitual diet during the study period." How was this measured, and what data support this claim? How much variation was there in the participant's diet at baseline? What about exercise regimen, sleeping cycle, or psychological stress, all of which are associated with gut microbiome changes? Erythromycin and azithromycin frequently cause nausea, and it appears that one participant in each treatment arm became nauseous and required the use of either prochlorperazine or metoclopramide. Surely, these two participants had at least some changes in their dietary habits if they became nauseous. Without blinding or placebo controls, a whole host of behavioral changes must be measured, precisely quantified, and modeled to show that these aren't confounding results. Unblinded studies will always be limited by unmeasured confounding.

2. Placebo effect: Placebo control has been shown to change systemic inflammation in studies of auto-immune disorders, including specific markers that the authors focused on, such as CRP (PMID: 32936297), and also influence host metabolism in other randomized trials (PMID:

36193169). Separating the antibiotic's impact from the placebo effect is impossible without a placebo control.

3. Regression to the mean: Some changes observed in the gut microbiome or inflammatory markers could be due to natural fluctuations over time rather than a direct effect of the antibiotics. Without a control group, it is impossible to determine the extent of this phenomenon.

4. That said, human data has merit, as it still has the advantage of temporality, and before/after comparisons can still be made. Still, this design and type of analysis are not quite at the evidence level to prove causality, and the authors should soften this language to “associated with.”

5. Causal claims can be made when comparing azithromycin to erythromycin. The fact that there was no significant difference in gut microbiome changes between the groups is reassuring validation for the author’s choice to study erythromycin alone for the murine model (given the inability to administer azithromycin through the water in mice). It is likely representative of azithromycin as well.

6. The supplemental table 1 describing the cohort should be moved to the main manuscript since part of its function is to address some of the limitations of confounding and bias and detect any measurable confounding between groups.

Minor concerns:

1. Consider using ellipses rather than “star patterns” of lines in the NMDS plots.

2. Just a point of clarification. In the supplemental methods, the authors state, “Faecal pellet collection before antibiotic treatment (20mg/kg erythromycin ethylsuccinate, supplemented in drinking water) and at day 90 was performed by placing mice into individual clean cages.” (supplemental methods, page 2, first paragraph in Mice Studies Section). Does this mean conventional mice were co-housed and only briefly removed and placed into individual clean cages for drug administration? I’m presuming that the authors took measures to control for the cage effect (PMID: 35545042), but please add a little more detail here to explain how the cage effect was accounted for. I think this is well described for germ-free mice in lines 440-448, but more detail would also be appreciated for the conventional mice.

Spectrum00831-23: The impact of long-term macrolide exposure on the gut microbiome and its implications for metabolic control

We thank the editorial team and the reviewers for their well-considered evaluation and comments on our manuscript. Below is a point-by-point response to the comments.

Reviewer comments:

Reviewer #1 (Comments for the Author):

Choo et al. conducted an intriguing translational study investigating the effects of macrolide-class antibiotics on host metabolism and inflammation. The study consisted of two parts. The first part was a small (20 subjects), randomized trial comparing the impact of azithromycin and erythromycin in healthy volunteers, which has its limitations, including the absence of placebo control, unblinded participants, and the concurrent use of anti-nausea medications in 10% of the population (Prochlorperazine in the erythromycin group and metoclopramide in the azithromycin group). In this part, the authors measured microbiome changes and markers of metabolism and inflammation. They performed an exploratory analysis of the changes in the microbiome's functional capacity associated with antibiotic treatment. They concluded that azithromycin and erythromycin altered the microbiome, host metabolism, and inflammation.

The second study was a murine model looking at erythromycin's influence on the gut microbiome, host metabolism, and host inflammation. The authors treated germ-free and conventional mice with erythromycin, measured changes in metabolism and inflammation, and found that erythromycin treatment caused impaired glucose tolerance and changes in inflammation. They then transplanted the fecal microbiome of antibiotic-treated and untreated mice into germ-free mice and found that fecal transplantation altered host metabolism but not inflammation. They concluded that changes in inflammation associated with erythromycin treatment directly affected the medication, while metabolic and inflammatory changes were mediated by the microbiome.

This is a compelling study. The murine model is well-designed, yielding fascinating results. However, I have significant concerns regarding the human randomized trial. The authors justify the lack of blinding and placebo control by stating, "This was felt acceptable as all outcomes were microbiological, not patient-orientated, and therefore not subject to bias." (Page 2, paragraph 1 in the supplement). I must express my strong disagreement with this assertion. The human data are an adjunct to the murine model and are informative but insufficient to establish causality. In my view, the study's real novelty is the murine model.

Major concerns:

1. Hawthorne effect: Since all participants in the human study knew their treatment status, they may modify their behavior accordingly, such as adhering to a healthier diet or exercising more, or any other variety of behavioral changes. These behavioral changes could

influence the gut microbiome metabolism and inflammatory markers independently of the antibiotics. The authors claim, "Participants maintained their habitual diet during the study period." How was this measured, and what data support this claim? How much variation was there in the participant's diet at baseline? What about exercise regimen, sleeping cycle, or psychological stress, all of which are associated with gut microbiome changes? Erythromycin and azithromycin frequently cause nausea, and it appears that one participant in each treatment arm became nauseous and required the use of either prochlorperazine or metoclopramide. Surely, these two participants had at least some changes in their dietary habits if they became nauseous. Without blinding or placebo controls, a whole host of behavioral changes must be measured, precisely quantified, and modeled to show that these aren't confounding results. Unblinded studies will always be limited by unmeasured confounding.

Response:

We thank the reviewer for their comprehensive review and constructive comments. We completely agree with the reviewer regarding the potential for each of these variables to influence the measures that we assessed. We should have been clearer on the respective rationales for the human and murine components of the study.

The primary aim of the initial analysis performed on human participants was to establish whether an association between macrolide exposure and changes in the gut microbiome and/or systemic homeostasis exists. Evidence of such phenomenon, at a clinically relevant macrolide dose, would support further investigation utilising a murine model (where potential confounders can also be controlled far more readily). Controlling for all potential confounders in humans (such as diet, exercise, sleep cycle and stress) would require a substantially larger cohort, as well as additional controls, including a placebo arm. Given the understandable reticence amongst recipients and prescribers to avoid antibiotic exposures that are not clinically directed, achieving this would have been challenging.

It is important to note the many potential confounding factors would be expected to influence participant microbiome characteristics in different ways, and that these exposures would vary between participants. However, the variations in microbiome characteristics observed following macrolide exposure were broadly conserved in both groups. Notwithstanding considerations relating to the inference of causality, we felt that these data provided sufficient rationale for exploring the effects of macrolide alteration of intestinal microbiology in a carefully designed murine model.

The concerns raised by the reviewer are valid and we have amended our manuscript to clarify the rationale for the study components and the appropriate interpretations that can be drawn from each (particularly relating to causation based on the cohort analysis). Ultimately, the findings that we present provide a rationale for now undertaking further investigations involving human participants, including those receiving long-term macrolides clinically.

2. Placebo effect: Placebo control has been shown to change systemic inflammation in studies of auto-immune disorders, including specific markers that the authors focused on, such as CRP (PMID: 32936297), and also influence host metabolism in other randomized trials (PMID: 36193169). Separating the antibiotic's impact from the placebo effect is impossible without a placebo control.

Response:

This is a valid point. We address this concern in our response to the previous comment, as well as in our responses below.

3. Regression to the mean: Some changes observed in the gut microbiome or inflammatory markers could be due to natural fluctuations over time rather than a direct effect of the antibiotics. Without a control group, it is impossible to determine the extent of this phenomenon.

Response:

We agree that the placebo control would potentially enable separation of natural variation from the effects of macrolide exposure. While longitudinal analysis reduces the impact of variation in host factors and diet, it does not exclude the influence of temporal variation.

Again, we would refer the reviewer to our response above relating to the rationale for this component of the study. We are careful not to suggest that causality can be attributed to macrolide exposure. Given that significant changes in microbiological and host markers were observed within participants following macrolide treatment, and that these variables did not significantly differ between participants in the azithromycin and erythromycin groups (at timepoints prior to, and after macrolide treatment), we feel that our analysis provided sufficient rationale for our detailed investigation in mice.

Based on our findings indicating causality in relationships observed in murine models, we now recommend the inclusion of a matched placebo control group in longitudinal assessment of larger human cohorts to explore the extent to which such causality exists and the relative influence of various potential confounders.

Line 401: *'Our findings suggest that ongoing macrolide therapy for chronic respiratory disease could result in significant direct impacts on systemic immunity and microbiome-mediated changes in metabolic homeostasis. Given that such effects have the potential to substantially influence long-term metabolic health outcomes, but may only become evident over long time period and in relation to diverse morbidities, longitudinal assessments of large cohorts, including matched placebo controls, are required.'*

4. That said, human data has merit, as it still has the advantage of temporality, and before/after comparisons can still be made. Still, this design and type of analysis are not quite at the evidence level to prove causality, and the authors should soften this language to "associated with."

Response:

We have now reviewed our manuscript and to ensure clarity around the limitations of the experimental design and the appropriate interpretation of the presented findings. These are reflected in the titles of each sections related to human studies (Line 93, 123 and 153) and the accompanying text under these sections and throughout the manuscript where appropriate.

5. Causal claims can be made when comparing azithromycin to erythromycin. The fact that there was no significant difference in gut microbiome changes between the groups is reassuring validation for the author's choice to study erythromycin alone for the murine model (given the inability to administer azithromycin through the water in mice). It is likely representative of azithromycin as well.

Response:

We agree and thank the reviewer for highlighting this.

6. The supplemental table 1 describing the cohort should be moved to the main manuscript since part of its function is to address some of the limitations of confounding and bias and detect any measurable confounding between groups.

Response:

We have now moved Supplemental Table 1 (Table S1) into the main manuscript as Table 1.

Minor concerns:

1. Consider using ellipses rather than "star patterns" of lines in the NMDS plots.

Response:

We thank the reviewer for the suggestion. We have revised the NMDS plots in Figure 2 and Figure 3 with ellipses.

2. Just a point of clarification. In the supplemental methods, the authors state, "Faecal pellet collection before antibiotic treatment (20mg/kg erythromycin ethylsuccinate, supplemented in drinking water) and at day 90 was performed by placing mice into individual clean cages." (supplemental methods, page 2, first paragraph in Mice Studies Section). Does this mean conventional mice were co-housed and only briefly removed and placed into individual clean cages for drug administration? I'm presuming that the authors took measures to control for the cage effect (PMID: 35545042), but please add a little more detail here to explain how the cage effect was accounted for. I think this is well described for germ-free mice in lines 440-448, but more detail would also be appreciated for the conventional mice.

Response:

We apologise for the lack of clarity. Conventional mice were housed between 3-5 mice per cage, with a total of six cages for each group (revision in line 454) in accordance with standard practice and ethical considerations (the importance of socialisation, etc). We now include statistical analysis that accounts for potential cage effects (revision in line 550), and report that microbiota differences remained significant.

Reviewer #2 (Comments for the Author):

This is an interesting study and meaningful for this research area. The manuscript was drafted well, then I only several main concerns as followings:

1. Please the authors clarify the rationals of sampling sizes used in the study.

Response:

Our exploration of macrolide-associated gut microbiome changes utilised samples from a study in which the primary goal was to assess acquisition of macrolide resistance in the oropharynx (PMID:35293783). The study required 10 participants in each (azithromycin and erythromycin) arm to obtain a type 1 error rate of 5% and a power of 0.9 (based on previously reported macrolide resistance induction rates from erythromycin (PMID: 23532242) and azithromycin studies (PMID: 23291349)). We assessed whether these sample sizes were sufficient for gut microbiome analysis by performing power calculation based on a previous study that assessed gut microbiome changes in children following a short-term (3-day) course of 10mg/kg azithromycin in children (PMID: 30478001). Based on the effect size calculated for changes in gut microbial richness and diversity of 0.90 and 0.64, respectively, a sample size of 13 or 23 participants is required for a one-tailed non-parametric paired analysis (type 1 error rate of 5% and power of 0.9).

To verify these sample sizes, we further determined the effect size based on a separate study that investigated gut microbiome changes 5 days after a single course of azithromycin treatment (20mg/kg) in children aged up to 60-months (PMID: 28402408). With an effect size of 0.73 for changes in microbial diversity, a sample size of 19 is required (type 1 error rate of 5% and power of 0.9). Therefore, we established that our sample size of 20 in this study will be sufficient to observe significant gut microbiome changes to assess their impact on host regulation.

We now include this information in supplemental material (Line 31):

'To further assess whether the sample sizes was sufficient for gut microbiome analysis, power calculation was also performed based on previous human studies that assessed macrolide-associated effects on the gut microbiome, which includes a short-term (3-day) course of 10mg/kg azithromycin in children², and a single course of 20mg/kg azithromycin treatment in children aged up to 60-months with gut microbiome analysis performed five days after the antibiotic treatment³. The effect size calculated for changes in gut microbial

diversity were 0.64 and 0.73, respectively. Therefore, a sample size of at least 19 is required to achieve a type 1 error rate of 5% and power of 0.9.'

2. Is it plausible to use parametric tests on microbial compositions comparisons?

Response:

Prior to statistical comparison of the groups, we assessed the variables in our microbiota dataset using the Shapiro-Wilk test to examine whether the data fit a normality distribution. Majority of the microbiota variables were non-parametric, which is common for microbial data (Weiss et al 2017; PMID: 28253908). We therefore used non-parametric tests for statistical analysis of the microbiome data.

3. What is the scale for those heatmaps in Figure 2.

Response:

We apologise for the oversight in including this information. The heatmap scale reflects the square root of taxa relative abundance, which is based on percentage levels (%). We have now included this information in Figure 2.

4. Legends of Figure 3 should be specified for different panels.

Response:

We thank the reviewer for pointing this. We have now specified the description for each panel in Figure 3:

'Figure 3. (A) *Non-metric multidimensional scaling (NMDS) ordination plot of bacterial functional pathway abundance in humans following treatment with azithromycin or erythromycin for 4 weeks. The ellipses of each group represent the standard deviation with 80% confidence limit (dotted and solid lines represent baseline and treatment groups, respectively).* (B) *Pairwise comparison between baseline and at the end of erythromycin (ERY) and azithromycin (AZM) treatment....'*

5. How the authors to exclude the confounding factors e.g. diet.

Response:

We thank the reviewer for raising a valid issue; the potential effect of confounding factors. We agree that external factors including diet can influence the gut microbiome and systemic homeostasis, including immune and metabolic regulation. We did not use a food diary to measure specific dietary components in this study, however, participants were asked to maintain their habitual diet during the study period (30 days). It is likely that participant's behavioural patterns will vary, including towards diet, resulting in potential alteration of the markers assessed in both directions, which can mask the effect of antibiotics. Given that we

observed directionality in microbiome and host marker changes, which were consistent for both azithromycin and erythromycin groups, these results suggests that while random effects may occur, stronger effects were driven by macrolide antibiotics.

Previous dietary intervention studies such as the CARDIVEG, which assessed the impact of a 3-month Mediterranean or vegetarian diet on the gut microbiome, indicated that the dietary intervention did not result in significant effects on the broad microbiota composition, although selected bacterial taxa were altered. Similar findings pertaining to the Mediterranean Diet and the gut microbiome were observed in a Men's Lifestyle Validation Study (n=307) (Wang et al 2021; PMID: 33574608). Based on these findings and the duration of our study, it is possible that confounding effects from diet would be modest. Our longitudinal study also utilised paired comparisons within subjects, which minimises the variability due to these confounding factors.

Furthermore, and as clarified in our responses to Reviewer 1, the principal goal of the analysis of the human cohorts was to establish whether a relationship between macrolide exposure and microbiome changes was evident, as a basis for further exploration in murine models where potential confounders can be readily controlled. We take care to emphasise that the observed relationships should be considered associative, given the inability to control for all potential confounders.

We now revised the limitations section to indicate the potential for confounding factors to contribute to effects on host physiology.

Line 417: *'However, as a placebo control group was not included, the contribution of natural variation in assessed markers over time, placebo effects on host physiological markers, or behavioural changes due to study participation (Hawthorne effect), to contribute or mask macrolide associated effects, cannot be determined.'*

5. All code used in the study is recommened strongly to deposit on github or other.

All code used in the study was derived from other studies and publicly accessible. We do not feel there is sufficient novelty in the manner in which we utilise this code to present it in relation to this work specifically. Instead, we have taken care to cite the original sources of the code used in each instance.

6. I also suggest authors to make a schematic figure to conclude the findings from the study at last.

Response:

We thank the reviewer for this suggestion and would be happy to take editorial guidance on this issue.

June 1, 2023

Dr. Jocelyn M Choo
South Australian Health and Medical Research Institute Limited
Adelaide, South Australia 5000
Australia

Re: Spectrum00831-23R1 (The impact of long-term macrolide exposure on the gut microbiome and its implications for metabolic control)

Dear Dr. Jocelyn M Choo:

Your manuscript has been accepted, and I am forwarding it to the ASM Journals Department for publication. You will be notified when your proofs are ready to be viewed.

Sincerely,

Ana Weil
Editor, Microbiology Spectrum
